# Molecular Characterization and Phylogenetic Analysis of Centipedegrass [*Eremochloa ophiuroides* (Munro) Hack.] Based on the Complete Chloroplast Genome Sequence

Haoran Wang [1,2], Yuan Zhang [1,2], Ling Zhang [1,2], Jingjing Wang [1,2], Hailin Guo [1,2], Junqin Zong [1,2], Jingbo Chen [1,2], Dandan Li [1,2], Ling Li [1,2], Jianxiu Liu [1,2] and Jianjian Li [1,2,*]

[1]  The National Forestry and Grassland Administration Engineering Research Center for Germplasm Innovation and Utilization of Warm-Season Turfgrasses, Institute of Botany, Jiangsu Province and Chinese Academy of Sciences, Nanjing Botanical Garden, Mem. Sun Yat-Sen, Nanjing 210014, China; njlydxwhr@163.com (H.W.); chenjb19@163.com (J.C.); dandan2007@163.com (D.L.)

[2]  Jiangsu Key Laboratory for the Research and Utilization of Plant Resources, Institute of Botany, Jiangsu Province and Chinese Academy of Sciences, Nanjing Botanical Garden, Mem. Sun Yat-Sen, Nanjing 210014, China

\*  Correspondence: lijianjian2021@yeah.net or lijianjian2015@cnbg.net; Tel.: +86-025-84347054; Fax: +86-025-84347072

**Abstract:** Centipedegrass (*Eremochloa ophiuroides*) is an important warm-season grass plant used as a turfgrass as well as pasture grass in tropical and subtropical regions, with wide application in land surface greening and soil conservation in South China and southern United States. In this study, the complete cp genome of *E. ophiuroides* was assembled using high-throughput Illumina sequencing technology. The circle pseudomolecule for *E. ophiuroides* cp genome is 139,107 bp in length, with a quadripartite structure consisting of a large single copyregion of 82,081 bp and a small single copy region of 12,566 bp separated by a pair of inverted repeat regions of 22,230 bp each. The overall A + T content of the whole genome is 61.60%, showing an asymmetric nucleotide composition. The genome encodes a total of 131 gene species, composed of 20 duplicated genes within the IR regions and 111 unique genes comprising 77 protein-coding genes, 30 transfer RNA genes, and 4 ribosome RNA genes. The complete cp genome sequence contains 51 long repeats and 197 simple sequence repeats, and a high degree of collinearity among *E. ophiuroide* and other Gramineae plants was disclosed. Phylogenetic analysis showed *E. ophiuroides*, together with the other two Eremochloa species, is closely related to *Mnesithea helferi* within the subtribe Rottboelliinae. These findings will be beneficial for the classification and identification of the *Eremochloa* taxa, phylogenetic resolution, novel gene discovery, and functional genomic studies for the genus *Eremochloa*.

**Keywords:** *Eremochloa ophiuroides*; chloroplast; genome structure; gene content; phylogeny

## 1. Introduction

Centipedegrass [*Eremochloa ophiuroides* (Munro) Hack.] is one of the most important $C_4$ perennial warm-season grass species that originated in China, and is mainly distributed in East Asia, South-east Asia, and the eastern and southern United States [1–3]. *E. ophiuroides* is mainly used as a turfgrass in tropical and subtropical regions and also as a pasture grass in some countries and regions of East Asia [4,5]. Now, it has been become an increasingly popular turfgrass due to its excellent adaptation to infertile acid soils, as well as for its resistance to the main biotic and abiotic stresses [2,6]. It also has great commercial application in land surface greening and soil conservation for its exceptional advantages of high ornamental value and low management and fertilization requirements [2,3]. Nevertheless, due to the serious reduction of wild grassland and wasteland areas caused by the rapid urbanization process in China, as well as the habitat destruction of the native grasses resulting from artificial farming and overgrazing, the natural populations of *E. ophiuroides*

have been experiencing a sharp decline. So, it is urgent to establish scientific strategies to protect and conserve the resources of *E. ophiuroides* in its main distribution areas.

Chloroplast (cp) is the key organelle of green plants responsible for photosynthesis and carbon fixation, and participates in the biosynthesis of a series of primary and secondary metabolites, such as amino acids, fatty acids, hormones, vitamins, nucleotides, and pigments [7,8]. It contains not only highly conserved genes essential to plant life but also more variable regions that are informative over broad time scales. Therefore, cp genome sequences can provide a valuable source for taxonomic studies and phylogenetic analysis among plant species and individuals [9,10], which could contribute greatly to plant breeding and conservation strategies. In addition, it is the characteristics of the non-recombinant nature, low mutation rates, and uniparental inheritance that make cp DNA significant in giving insights into plant evolution and developing applications for biotechnological breeding [11,12]. At present, the rapid development of high-throughput sequencing technology makes it convenient and inexpensive to assemble plant cp genomes and implement whole genome-based phylogenomics [13]. In contrast to previous studies done with a single or a few cp loci-based approaches, using the complete cp genome information now provides a unique opportunity to investigate related species evolution based on whole-genome comparison [14,15].

As an important turfgrass, the turf quality of *E. ophiuroides*, to a great extent, depends on its biomass, color, and green color retention, which are highly correlated with photosynthetic efficiency. Like most other turf grass species, *E. ophiuroides* prefers to grow in open and sunny places. Therefore, exploring chloroplast genes related to photosynthesis could contribute to breeding in turf grasses [16]. Analyzing and characterizing the cp genome of a turf grass would provide invaluable information to improve the turf quality and also to facilitate the development of a plastid transformation system for the plant [17]. However, despite being the most popular turfgrass introduced into the USA by Frank Meyer one century ago [18], the research on *E. ophiuroides* cp genome is lagging behind. To date, there is no relevant report to interpret *E. ophiuroides* cp genome in detail, which is not conducive to our understanding and progress of *E. ophiuroides* evolution, species identification, germplasm conservation, genetic engineering, and other related research.

In the present study, we sequenced the *E. ophiuroides* cp genome using Illumina technology, assembled the complete cp genome sequence of *E. ophiuroides*, characterized the cp genomic structure, and performed detailed phylogenetic analyses using complete cp genome sequence information. We also analyzed the fully assembled cp genome of *E. ophiuroides* and compared it to seven related species of Gramineae. The main purposes of this study were to investigate the cp genome structure of the *E. ophiuroides*, to explore the phylogenetic position of *E. ophiuroides* in the tribe Andropogonodae, and also to provide basic data for further molecular studies related to the identification and phylogenetic classification of Eremochloa species, chloroplast genes discovery, and functional genomic studies in the genus *Eremochloa*.

## 2. Results

### 2.1. Genome Assembly and Structure Analysis

A total of 19,101,863 clean reads (approximately 5.73 Gb) were obtained from the *E. ophiuroides* leaf library. After reference-guided denovo assembly of the reads with minor modifications, a complete circular pseudomolecule was generated for *E. ophiuroides* cp genome with a total length of 139,107 bp (GeneBank accession: MT806102). Since 738,323 reads were mapped to the assembled cp genome, the sequencing depth for the cp genome reached more than 1500× (Figure 1, Table 1).

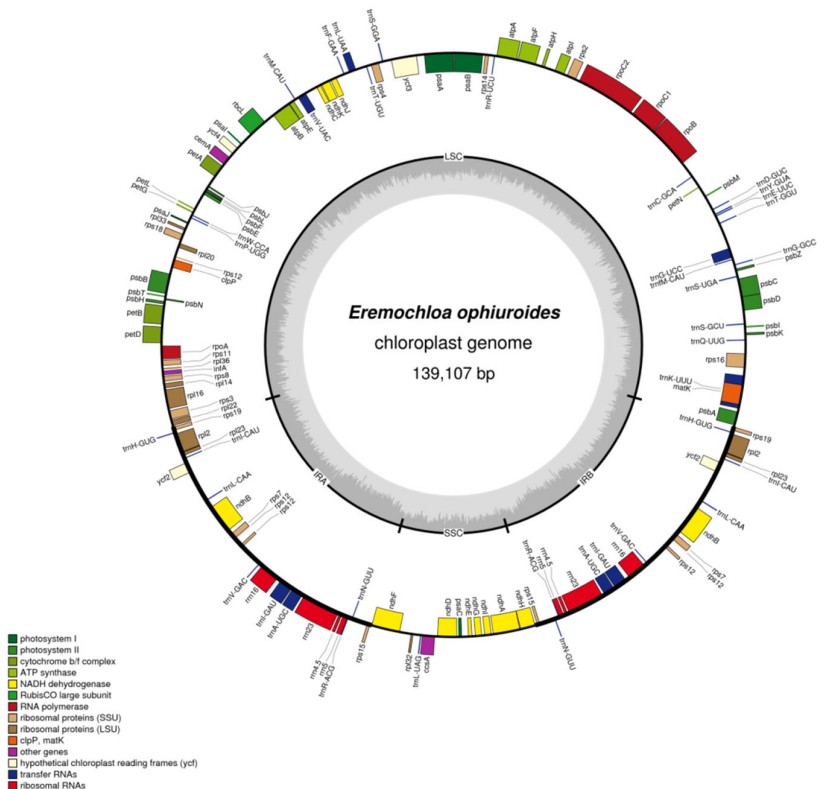

**Figure 1.** Circular gene map of *E. ophiuroides* complete chloroplast genome. Genes shown on the outside of the circle are transcribed clockwise, and genes inside the circle are transcribed counter-clockwise. Genes belonging to the same functional groups are color-coded. The darker gray and lighter gray in the inner circle indicates the GC content and AT content, respectively.

**Table 1.** Chloroplast genome features in *Eremochloa ophiuroides*.

|  | T/U (%) | C (%) | A (%) | G (%) | Length (bp) | AT (%) |
|---|---|---|---|---|---|---|
| Genome | 30.75 | 19.15 | 30.86 | 19.25 | 139,107 | 61.60 |
| LSC | 32.12 | 17.94 | 31.64 | 18.3 | 82,081 | 63.76 |
| SSC | 31.49 | 16.97 | 35.83 | 15.71 | 12,566 | 67.32 |
| IRa | 28.03 | 23.01 | 27.97 | 20.99 | 22,230 | 56.00 |
| IRb | 27.97 | 28.03 | 28.03 | 23.01 | 22,230 | 56.00 |

The cp genome of *E. ophiuroides* exhibited a typical circular quadripartite structure, composed of a large single-copy (LSC) region of 82,081 bp with 63.76% AT and a small single-copy (SSC) region of 12,566 bp with 67.32% AT separated by a pair of inverted repeat (IR) regions of 22,230 bp each with 56.00% AT (Figure 1). The genomic AT content in the *E. ophiuroides* cp was 61.60% (Table 1).

*2.2. Gene Annotation*

A total of 131 genes were annotated in the *E. ophiuroides* cp genome, of which 20 genes are duplicated in the IR regions and 111 are unique, including 77 protein-coding genes, 30 tRNA genes, and 4 rRNA genes (Figure 1, Table 2). Except for each of the six tRNA genes and eight protein-coding genes having one intron, and each of two protein-coding genes containing two introns, most of the unique genes were identified with no introns. Gene function analysis revealed that all the unique genes could be classified into four categories, including genes associated with photosynthesis (44 genes), genes involved in self-replication (59 genes), genes with other functions (5 genes), and genes of unknown function (3 genes) (Table 2).

**Table 2.** Summary of gene annotation for the *E. ophiuroide* cp genome.

| Category | Gene Group | Gene Name |
|---|---|---|
| Photosynthesis | Subunits of photosystem I | psaA, psaB, psaC, psaI, psaJ |
| | Subunits of photosystem II | psbA, psbB, psbC, psbD, psbE, psbF, psbH, psbI, psbJ, psbK, psbL, psbM, psbN, psbT, psbZ |
| | Subunits of NADH dehydrogenase | ndhA *, ndhB *(2), ndhC, ndhD, ndhE, ndhF, ndhG, ndhH, ndhI, ndhJ, ndhK |
| | Subunits of cytochrome b/f complex | petA, petB *, petD *, petG, petL, petN |
| | Subunits of ATP synthase | atpA, atpB, atpE, atpF *, atpH, atpI |
| | Large subunit of rubisco | rbcL |
| Self-replication | Proteins of large ribosomal subunit | rpl14, rpl16 *, rpl2 *(2), rpl20, rpl22, rpl23(2), rpl32, rpl33, rpl36 |
| | Proteins of small ribosomal subunit | rps11, rps12 **(2), rps14, rps15(2), rps16 *, rps18, rps19(2), rps2, rps3, rps4, rps7(2), rps8 |
| | Subunits of RNA polymerase | rpoA, rpoB, rpoC1, rpoC2 |
| | Ribosomal RNAs | rrn16(2), rrn23(2), rrn4.5(2), rrn5(2) |
| | Transfer RNAs | trnA-UGC *(2), trnC-GCA, trnD-GUC, trnE-UUC, trnF-GAA, trnG-GCC, trnG-UCC *, trnH-GUG(2), trnI-CAU(2), trnI-GAU *(2), trnK-UUU *, trnL-CAA(2), trnL-UAA *, trnL-UAG, trnM-CAU, trnN-GUU(2), trnP-UGG, trnQ-UUG, trnR-ACG(2), trnR-UCU, trnS-GCU, trnS-GGA, trnS-UGA, trnT-GGU, trnT-UGU, trnV-GAC(2), trnV-UAC *, trnW-CCA, trnY-GUA, trnfM-CAU |
| Other genes | Maturase | matK |
| | Protease | clpP |
| | Envelope membrane protein | cemA |
| | c-type cytochrome synthesis gene | ccsA |
| | Translation initiation factor | infA |
| Genes of unknown function | Conserved hypothetical chloroplast ORF | ycf2(2), ycf3 **, ycf4 |

* Gene with one introns, ** Gene with two introns, (2) Number of copies of multi-copy genes.

*2.3. Codon Preference Analysis*

In addition to the 3 termination codons, there were 63 codons encoding 20 diverse amino acids in *E. ophiuroides* cp genes (Figure 2, Table S1–see Supplementary Materials), and almost half of the codons encoding amino acids had codon preferences. A total of 31 codon preferences were identified from all the codons, of which 30 encode 18 amino acids, and one is the termination codon. According to the partitions of synonymous codon preferences, 70.97%, 12.90%, and 16.13% of the preferred codons displayed high (RSCU value > 1.3), moderate ($1.2 \leq$ RSCU value $\leq 1.3$),) and low preferences ($1.0 <$ RSCU value $< 1.2$), respectively.

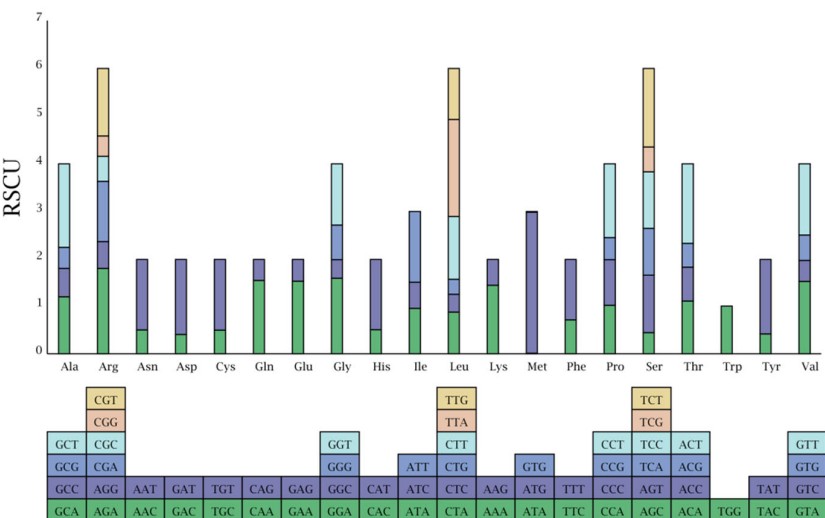

**Figure 2.** Codon content of 20 amino acids in all cp protein-coding genes of the *E. ophiuroides*.

### 2.4. Repeat Structure and SSR Analysis

Repeated analysis revealed 20 palindromic repeats, 29 forward repeats, and two reverse repeats in the cp genome of the *E. ophiuroides* (Table S2). The forward repeat units were 30–242 bp long, and most of the forward repeats were positioned in the LSC regions except for four located in IR regions and one in the SSC region. Similar to the forward repeats, the majority of palindromic repeat units were 30–242 bp in length and distributed in LSC regions, with one of them being 22,230 bp long as an exception. Alternatively, for the reverse repeats, two in total were less than 35 bp in length and were identified in LSC regions.

Microsatellite repeats were also extracted in *E. ophiuroides* genome (Table S3). A total of 197 simple sequence repeats (SSRs) loci were identified, including 191 perfect SSRs and six complex SSRs. Among these perfect SSRs, 129 (67.54%) SSRs were mononucleotide repeats, and the rest (in descending order of abundance) were tri- (46, 24.08%), tetra- (10, 5.24%), di- (5, 2.62%), and penta- (1, 0.52%) nucleotide repeats. Most of the mononucleotide repeats were composed of poly A (43.41%) and poly T (50.39%) repeats, whereas only a few mononucleotides were poly G and poly C repeats (6.20%). For dinucleotides, four out of five were AT/TA repeat motifs, and the rest was a TC repeat. Among the trinucleotide and tetranucleotide repeat motifs, the proportion of A or T in trimers and tetramers reached 71.01% and 55.00%, respectively. One pentanucleotide was an ATAAA repeat.

### 2.5. Expansion and Contraction of IR and Genome Collinearity

The exact positions of IR boundaries and their adjacent genes of the *E. ophiuroides* were compared with the other seven species from the family Gramineae or Poaceae (Figure 3). In the cp genome of the three Eremochloa species, the IR boundary positions and their adjacent genes were exactly the same. IRa/SSC and IRb/SSC junctions were found within the gene ndhH and the gene ndhF, respectively, and correspondingly the ndhH pseudogene (1 bp), the ndhF pseudogene (29 bp) was observed at the IRa/SSC boundary and the IRb/SSC border, whereas no pseudogene was detected at the IRa/LSC and IRb/LSC boundaries. As for Sorghum and Zea, they were found to have exactly the same IR boundary position and the adjacent genes and to have almost unanimous IR situations with Eremochloa only, except for the positions of the genes rpl22 (57 bp to IRb) and psbA (88 bp to IRa) in LSC. Setaria italic had IR boundary positions similar to that of Sorghum bicolor or *Zea mays*. IR boundary positions and the adjacent genes of Oryza sativa and Brachypodium distachyon were generally consistent with that of Eremochloa plants, even though slight differences were observed, such as the position divergence of the gene ndhF adjacent to IRb/SSC junction, the pseudogene ndhH length variations detected in IRa regions of the two species.

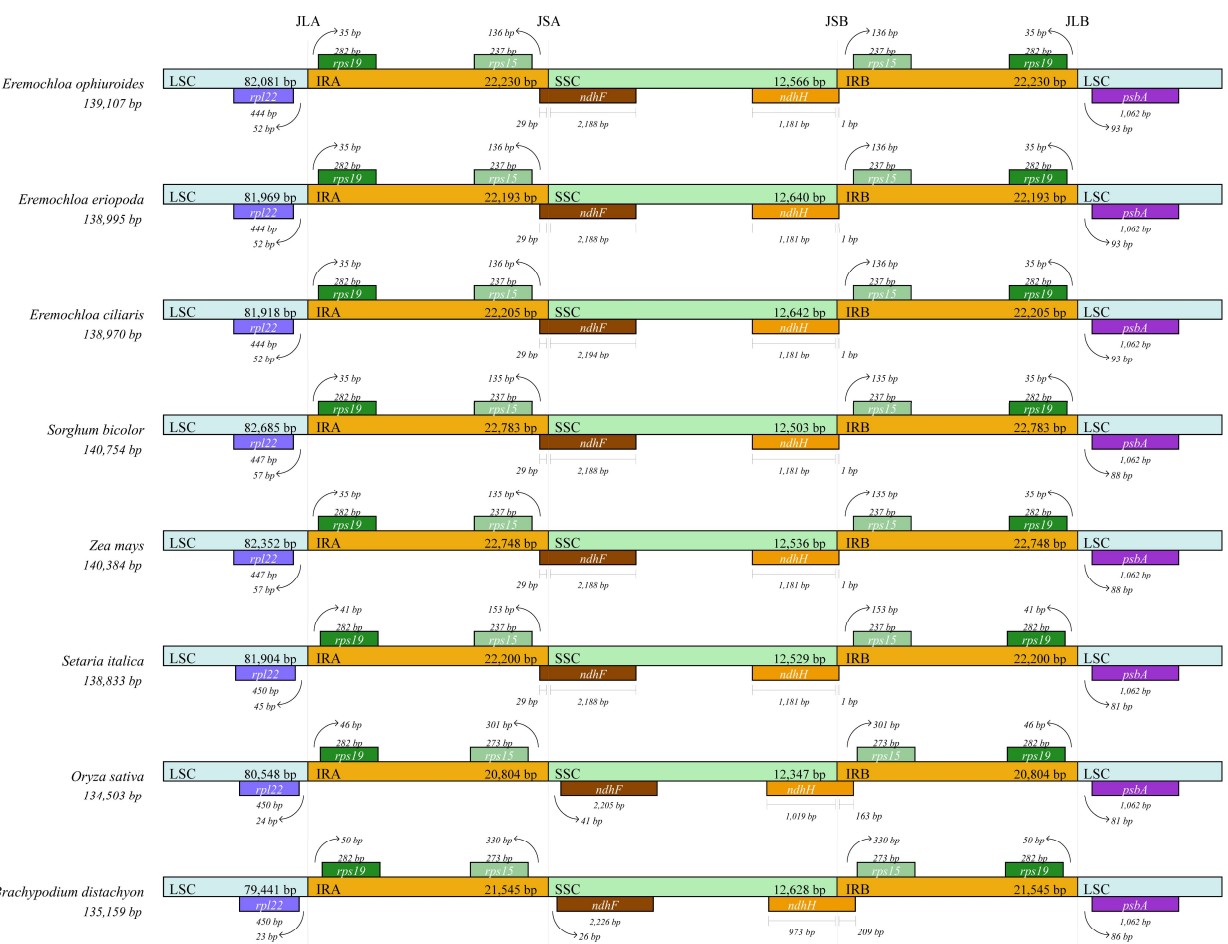

**Figure 3.** Comparison of the borders of LSC, SSC, and IR regions among eight cp genomes. The regions of LSC, IR, and SSC were indicated using three different colors, respectively. JLA: junction line between LSC and IRA; JSA: junction line between SSC and IRA; JSB: junction line between SSC and IRB; JLB: junction line between LSC and IRB.

Collinearity analyses revealed that all cp genomes of the eight species formed locally collinear blocks (LCBs). The gene order of the three Eremochloa cp genomes was especially highly conserved compared with that of other plant species (Figure 4). Although the highest cp genome homologies were detected between *Eremochloa* and *Sorghum*, the order of cp gene loci was highly consistent among the Gramineae plant genomes. This demonstrates that the cp genome has a high homology among Gramineae plants.

### 2.6. Phylogenetic Relationship

Phylogenetic relationships of species in the tribe Andropogoneae and taxonomic statuses of *E. ophiuroides* and other species in the same tribe were systematically classified through maximal likelihood (ML) analysis of the newly sequenced and published complete cp sequences. Forty-nine published or available complete cp genome sequences and a newly sequenced *E. ophiuroides* cp sequence were combined in this study. Thus, we reconstructed a phylogenetic tree of the tribe Andropogoneae using a total of 50 cp genomes, which were selected from 46 different species in Andropogoneae (three species with two cp sequences) and one species (*Arundinella deppeana*) in Arundinelleae used as an outgroup (Table S4). The newly constructed phylogenetic tree fully supported that *E. ophiuroides* is closely related to *Eremochloa ciliaris* and *Eremochloa eriopoda* with 100% bootstrap values. The three *Eremochloa* species, together with *Mnesithea helferi*, form one monophyletic group corresponding to the subtribe Rottboelliinae (Figure 5).

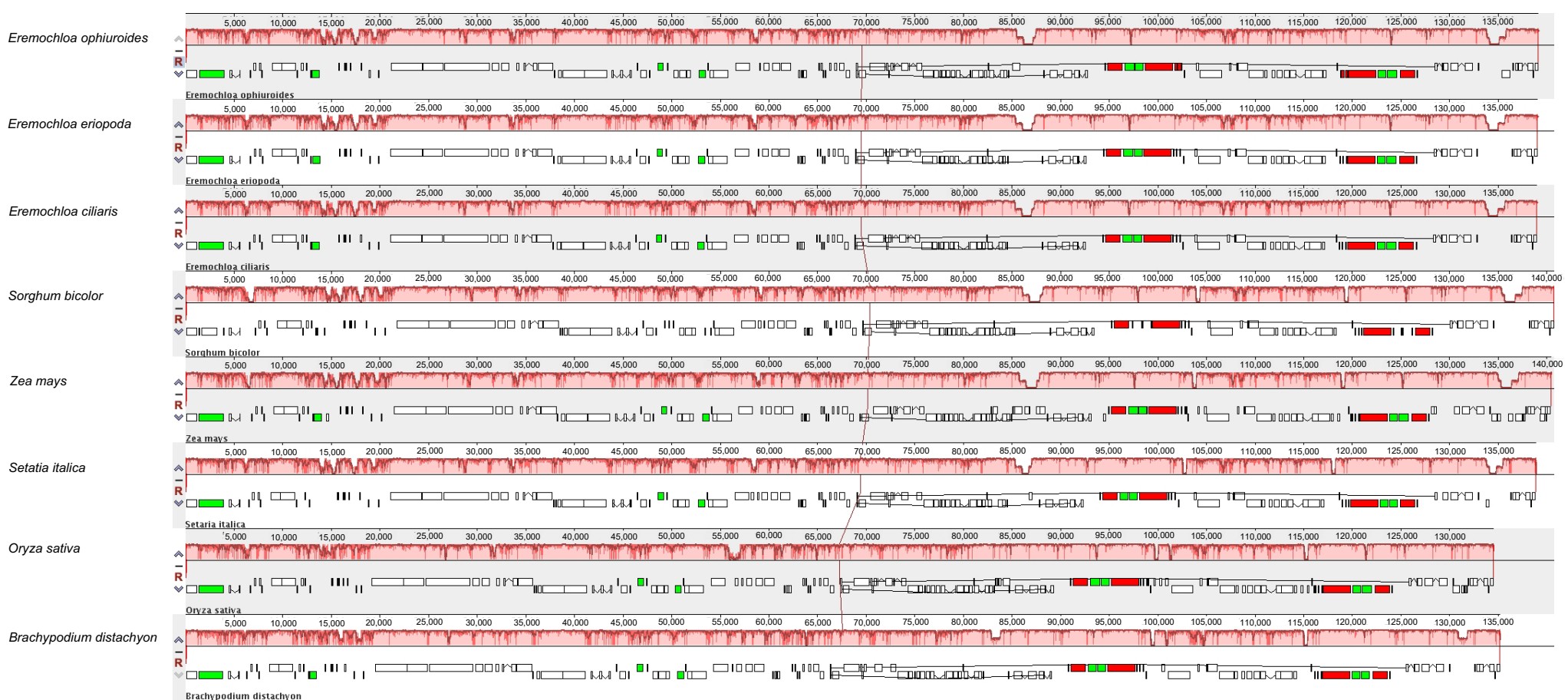

**Figure 4.** Comparison gene collinearity analysis of the chloroplast genomes from eight Gramineae species. Annotations of protein-coding genes, tRNA genes, and rRNA genes are indicated using white, green, and red boxes, respectively.

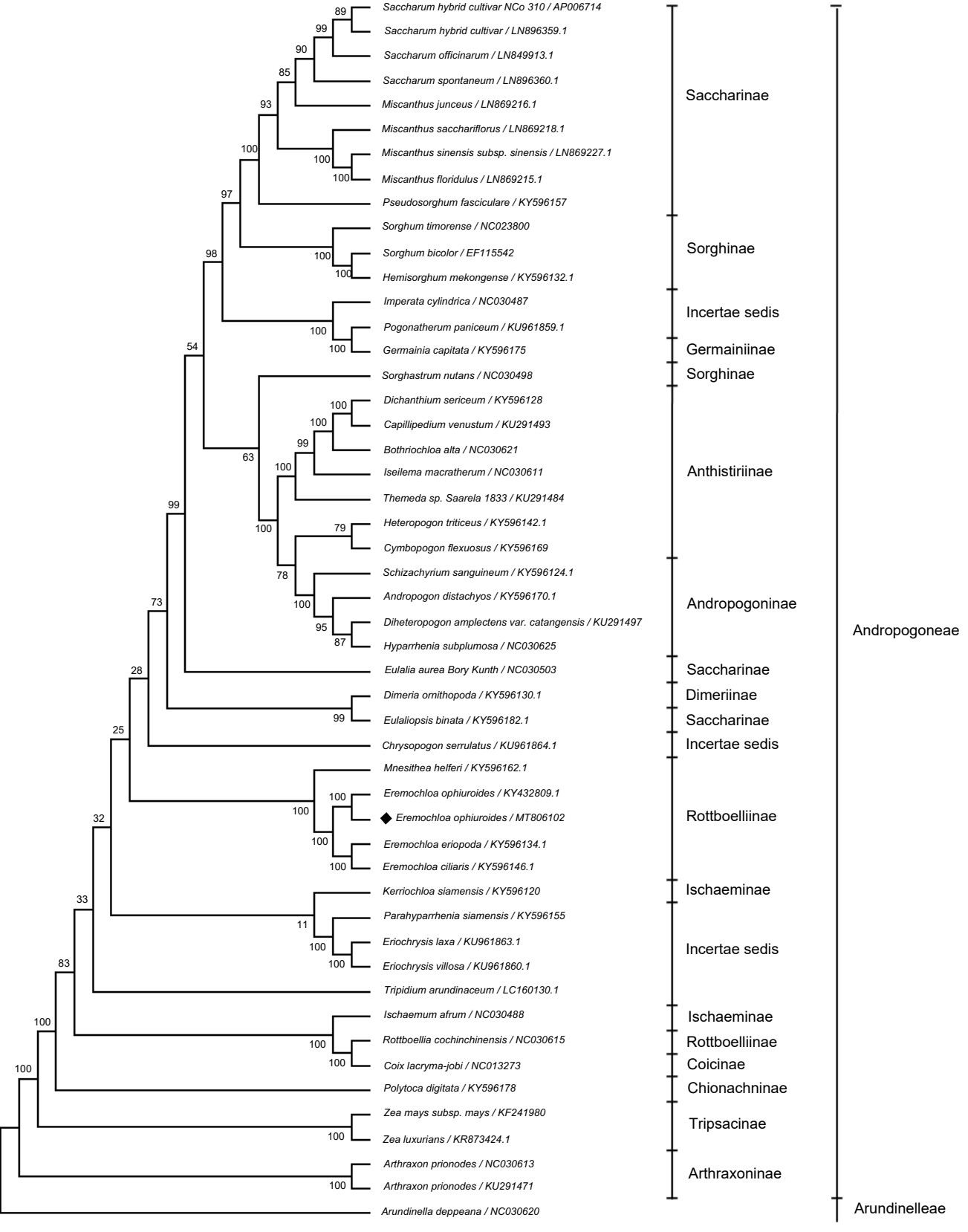

**Figure 5.** Phylogenetic tree of the Andropogoneae species based on the complete cp genome data by maximum likelihood (ML). A total of 50 species were used to reconstruct a phylogenetic tree using MEGA6 software (version 6.0), and *Arundinella deppeana* was used as the outgroup. Subtribes and higher taxonomic groupings are indicated.

It is noteworthy that a larger number of species traditionally classified as the same subtribe do not form a group. From the phylogenetic analysis of this study, only four mono­phyletic groups, corresponding to subtribe Saccjaromae, Sorghinae, Andropogoninae, and Rottboelliinae, can be retrieved from the tree, while quite a few non-monophyly are formed. A typical instance is the placement of *Germainia capitata* (Germainiinae) as a sister to *Pogonatherum paniceum* (Incertae sedis) with the same branch length. Similar cases can be found for the placement of *Dimeria ornithopoda* (Dimeriinae) as the sister of *Eulaliopsis binata* (Saccharinae) and the placement of *Rottboellia cochinchinensis* (Rottboelliinae) as a sister of *Coix lacryma-jobi* (Coicinae). In addition, *Heteropogon triticeus* and *Cymbopogon flexuosus*, two species in Anthistiriinae, were clustered to a sister clade of Andropogoninae; *Kerriochloa siamensis*, one species of Ischaeminae, was constrained as sister to Incertae sedis.

## 3. Discussion

### 3.1. Genome Size and Gene Identification

The *E. ophiuroides* cp genome is 139,107 bp in length, which is similar to those cp genomes in the Panicoideae subfamily, which range from 138 Kb in *Setaria viridis* [19] to 141 Kb in *Saccharum offcinarum* [20] but is larger than those of species in Chloridoideae, Pooideae, and Oryzoideae subfamilies with not more than 137 Kb (Table S5). Judged from the average size of ~137 kb for most Poaceae cp genomes [21], *E. ophiuroides* cp genome is of average size within Panicoideae and of large size within Poaceae (Gramineae). The cp pseudomolecule of *E. ophiuroides*, like that of most angiosperms, is circular with a typical quadripartite structure with LSC and SSC regions separated by a pair of IRs. The overall AT content of the *E. ophiuroides* cp genome was 61.6%, which is similar to that of most Gramineae plants (~61%, Table S5).

The gene and intron contents in the *E. ophiuroides* cp DNA are basically identical to those of rice [22,23], wheat [24], maize [25], sorghum [26], and other grass species [20,21,27,28], with 77 protein-coding genes, 30 tRNA genes, and 4 rRNA genes. Among the 111 unique genes, 1 intron was detected in the 14 genes (6 tRNA and 8 protein-coding genes) and 2 introns were in the 2 genes (rps12 and ycf3). For all identified genes, 59 fragments are related to self-replication, and 44 genes are associated with photosynthesis. Of the 44 photosynthesis-related genes, five genes (*psaA, B, C, I, J*) are related to photosystem I, 15 genes encode photosystem II component, and six genes (*atpA, B, E, F, H*, and *I*) are responsible for ATP synthase and the other 18 genes encode the components of the electron transport chain. Similar characteristics of the protein-coding genes were also present in a few *Oryza* species [23,29,30].

### 3.2. Repeat Sequence

The nucleotide sequences of most organism genomes contain many different types of repetitive sequences, such as short tandem repeats, interspersed repeats, or spaced repeats. These repeat elements are either dispersed throughout the genome or within a short region of the genome [31]. The mismatch and false recombination of the sequences generally result in the occurrence of sequence variation and DNA rearrangement [32,33]. In the present study, quite a few long repeats, including forward and palindromic repeats and reverse repeats, were detected in *E. ophiuroides* cp genome sequences, and most of them were distributed in LSC regions of the genome. Similar results were also observed in other plant species, such as *Swertia mussotii* [34] and *Oryza minuta* [23]. This reflects the common characteristics of the LSCs in most plant cp genomes.

SSRs, also called microsatellites, are known to be more informative and are very abundant and evenly distributed in angiosperm plastomes [35]. Because of their high abundance and polymorphism, and ubiquitous distribution, SSRs have been extensively used as versatile DNA markers in plant genetic and genomic studies [36]. SSR length and abundance, as well as the motif type, are the key characteristics of microsatellites [37]. Besides complex SSRs, five types of perfect SSRs (mono-/di-/tri-/tetra-/penta-nucleotide repeats) were found in the *E. ophiuroides* cp genome sequences. Mononucleotide repeats were the most abundant cp SSR motif, followed by trinucleotide and tetranucleotide

repeats in detected *E. ophiuroides* cp SSRs. This result is not completely consistent with other findings that showed either mono- or dinucleotides are the most frequent SSR type in many plant cp genomes [38–40], but is consistent with the reports in Lythraceae species [41] and *Magnolia polytepala* [12], and is also in accord with the finding of SSR mining from the *E. ophiuroides* RNA-seq data although mononucleotide repeat was omitted in that study [42]. No matter which type of SSRs were detected in the present study, most of them were rich in AT content. This is congruent with some reported chloroplast SSRs [43–45].

### 3.3. IR Expansion and Contraction

A prominent feature of cp genomes is the presence of IRs. Expansion and contraction of IR region boundaries generally lead to size variations in the cp genome, which plays a significant role in species evolution [46]. In this study, a detailed analysis of IR expansion and contraction in cp genomes of *E. ophiuroides* and seven other species was conducted by comparing the positions of four junctions, i.e., JLA (junction line between LSC and IRa), JSA (junction line between SSC and IRa), JLB (junction line between LSC and IRb), JSB (junction line between SSC and IRb), and the lengths of two IRs (IRa and IRb) and two single-copy regions (LSC and SSC), and exact IR border positions and adjacent genes as well. The IR region of *E. ophiuroides* was 22,230 bp in length, which was a medium length in the eight compared species from 20,804 bp to 22,783 bp. JLA is between *rps19* and *rpl22*, and JLB is located between *rps19* and *psbA* in all eight Gramineae species. Both the distances between *rps19* and JLA, between *rps19* and JLB are 35 bp in all three Eremochloa species, *S. bicolor* and *Z. mays*, which are shorter than that in other three Gramineae species; the distance between *rpl22* and JLA in three Eremochloa species is shorter than that in *S. bicolor* and *Z. mays*, but is longer than that in the other species, while the distance between *psbA* and JLB in three Eremochloa species is longer than that in the other Gramineae species. The *ndhF* gene traverses the SSC and IRa regions, with 29 bp located in the IRa region for all the $C_4$ plants, for which photosynthetic enzymes are located in the chloroplasts of the mesophyll and/or bundle sheath cells, including three Eremochloa species, *S. bicolor*, *Z. mays* and *S. italic*, but is located in the SSC region for $C_3$ plants, for which all photosynthetic enzymes are confined in the bundle sheath cells, of *O. sativa* and *B. distachyon*, as revealed in the present study, or of *O. minuta*, as reported by Asaf et al. [23]. Although the *ndhH* gene stretched across the SSC and IRb regions, approximately 1181 bp were located in the IR region, while only 1 bp was in the IRb region for all species except for *O. sativa* and *B. distachyon*. This is in accord with most reported findings in Gramineae plants [23]. This hints that variation in the JSA border caused by IR expansion or contraction might result in the difference between $C_3$ and $C_4$ plant cp genomes. Our results also demonstrated that size variation of cp genomes resulting from IR contraction and expansion is a common feature during the evolution of Gramineae plants, although structural organization and gene order of Gramineae cp genomes are highly conserved [47].

### 3.4. Phylogenetic Analysis

The tribe Andropogoneae includes over 1200 species in ca. 90 genera, and is a primary component of grasslands and savannahs throughout most of the world's tropical and subtropical regions [48,49]. Recently, a number of phylogenetic and evolutionary studies have been implemented for the tribe Andropogonodae using complete chloroplast genomes [49–52]. Although *E. ophiuroides* is an important member of the genera Eremochloa of the tribe Andropogoneae, it has not been included in these studies, which restricts illuminating its evolutionary relationships to other Andropogoneae species. Our molecular phylogenetic tree based on sequences of complete cp genomes revealed a close relationship between *E. ophiuroides*, *E. ciliaris*, and *E. eriopoda*, and their placement in a common clade with *Mnesithea helferi* is highly supported with bootstrap values of 100% within the subtribe Rottboelliinae (Figure 5). This is congruent with the traditional morphology-based taxa of Rottboelliinae, indicating that the classification of subtribe Rottboelliinae is generally reasonable.

In addition, from our results, the Rottboelliinae, Saccjaromae, Sorghinae, and Andropogoninae are typically monophyletic groups, which reflect the agreement between molecular phylogeny and traditional morphology-based taxonomy. However, some non-monophylies of subtribes were recognized in the current molecular phylogeny. In the present study, *Germainia capitata* (Germainiinae) was placed as a sister to *Pogonatherum paniceum* (Incertae sedis), *Dimeria ornithopoda* (Dimeriinae) as a sister to *Eulaliopsis binata* (Saccharinae), and *Rottboellia cochinchinensis* (Rottboelliinae) as a sister to *Coix lacryma-jobi* (Coicinae), which are congruent with previous results for these species [49–52]. Another typical non-monophyletic area in the tree is the placement of *Heteropogon triticeus* and *Cymbopogon flexuosus* (two species in Anthistiriinae) in a common clade with Andropogoninae species, and a similar result has actually been reported [49]. However, it is worth mentioning that *Sorghastrum nutans* and *Eulalia aurea* were not clustered as sister clades in the current study, which is incongruent with previously reported results [50–52]. This is mainly due to the fact that more extensive species (50 complete cp genome data of 47 different species) in the tribe Andropogoneae were used for phylogenetic analysis in the present study. Considering the same monophyletic clades clustered between *Rottboellia cochinchinensis* and *Coix lacryma-jobi*, *Germainia capitata* and *Pogonatherum paniceum*, *Dimeria ornithopoda* and *Eulaliopsis binata*, and the different monophyletic clades formed from *Eulalia aurea* and *Eulaliopsis binata* displayed in this study, combined with previously reported phylogenetic relationships between these species [50–52], future more sampling with better balancing of ingroup Rottboelliinae, Coicinae, Germainiinae, Incertae sedis, Dimeriinae, Saccharinae should be considered so as to better address questions of subtribal monophylies.

## 4. Materials and Methods

### 4.1. Plant Material, DNA Extraction and Sequencing

The experimental material was an elite *E. ophiuroides* cultivar "Ganbei", which was originally collected from Lushan, Jiangxi province, and now is deposited in the nursery of the National Main Warm-season Turfgrass Gene Bank (NMWTGB) at the Institute of Botany, Jiangsu Province and Chinese Academy of Sciences, Nanjing Botanical Garden, Men. Sun Yat-sen, China, with an accession No. E039 in NMWTGB. Total genomic DNA was extracted from fresh young leaves of E039 using the EZgene^TM SuperFast Plant Leaves DNA Kit (Biomiga, San Diego, CA, USA) following the manufacturer's protocol. The quality and integrity of the DNA were checked and determined using spectrophotometry and agarose gel electrophoresis, respectively. The high-quality DNA was then divided into 300~500 bp fragments using an ultrasonicator (Covaris M220, Covaris, Woburn, MA, USA). Average 350 bp paired-end (PE) libraries were prepared using Illumina TruSeq DNA Sample Prep kit (Illumina Inc., San Diego, CA, USA) and were then sequenced on Illumina's *NovaSeq* 6000 platform.

### 4.2. Data Assembly, Gene Annotation and Codon Preference Analysis

Raw reads were filtered using the base quality control software fastp (version 0.20.0) to obtain high-quality reads. Then a BLAST analysis was performed between the high-quality reads and the reference cp genome (KY432809.1) to extract cp-like reads. The obtained high-quality cp-like reads were further assembled into contigs via de novo assembler SPAdes v3.9.0 [53]. Then a draft sequence was generated by integrating all contigs using NOVO-Plasty with the reference genome (KY432809.1) as a template. Cp cyclization and initiation site determination was done by manual processing. Prodigal [54] and hmmer [55] software were applied to annotate protein-coding genes and ribosomal RNAs, respectively, while transfer RNAs were predicted via Aragorn software (version 1.2.41) [56]. The annotation results were verified using the CpGAVAS pipeline and then manually corrected. Finally, the genome map was constructed using the OrganellarGenomeDRAW tool (OGDRAW) [57].

The codon preference was analyzed by R software (version 4.1.3). The degree of the codon preference was evaluated by the relative synonymous codon usage (RSCU). The RSCU value was computed as the ratio between the use frequency and the expected

frequency of a particular codon. According to the RSCU theory [58,59], synonymous codon preference was artificially categorized into four models: high preference (RSCU > 1.3), moderate preference (1.2 ≤ RSCU ≤ 1.3), low preference (1.0 < RSCU < 1.2), and no preference (RSCU ≤ 1.0).

### 4.3. Comparative Analysis and Repeat Sequence Identification

Besides the newly sequenced *E. ophiuroides* cp genome, the complete cp genome sequences of seven different plant species, including *Eremochloa ciliaris* (KY596146.1), *Eremochloa eriopoda* (KY596134.1), *Sorghum bicolor* (EF115542.1), *Zea mays* (AY928077.1), *Setaria italic* (KJ001642.1), *Oryza sativa* (MG252500.1), and *Brachypodium distachyon* (KU170609.1), were downloaded from NCBI Organelle Genome Resources database. Collinearity analysis among the eight species was implemented by using Mauve (http://darlinglab.org/mauve, accessed on 25 July 2021) software with default parameters.

Tandem repeats in the cp genome were assessed using the Tandem Repeat Finder program [60] with default settings. Forward, palindromic, reverse, and complement repeats were detected using vmatch v2.3.0 with the minimal repeat size setting greater than 30 bp and a Hamming distance of 3. Simple sequence repeats (SSRs) or microsatellites were identified using the Perl script MISA v1.0 [61], and the parameters were set for identifying mono-, di-, tri-, tetra-, penta- and hexa-nucleotide motifs with a threshold of eight, five, three, three, three, and three repeat units, respectively; microsatellites with interruption bases between two adjacent microsatellites or two different microsatellites connected directly are specified as complex SSR.

### 4.4. Phylogenetic Analysis

The phylogenetic analysis was conducted based on *E. ophiuroides* cp genome data including a newly sequenced cp genome (MT806102) in the present study and another one submitted to NCBI GeneBank (KY432809.1) by Gallaher et al. in 2017, together with the cp genomes of 48 Andropogonodae species with strong genetic relationships downloaded from GeneBank, and *Arundinella deppeana* (NC030620) was selected as the outgroup (Table S4). MAFFT [62] and trimAl [63] were used for genome sequence alignment and data set trimming, respectively. The best substitution model (GTR+G) was applied as suggested by jModelTest v2.1.7 [64], and the randomized axelerated maximum likelihood (RAxML) program was chosen to perform a phylogenetic analysis with 1000 bootstrap replicates in MEGA 6.0.

### 5. Conclusions

The complete cp genome sequence of *E. ophiuroides* (139,107 bp) was present in this study. The circular cp genome of *E. ophiuroides* possesses a typical quadripartite structure, which is well conserved among most cp genomes from the Gramineae family. We annotated a total of 131 genes in the *E. ophiuroides* cp genome and found that 44 of the genes are involved in photosynthesis. Most *E. ophiuroides* codons encoding amino acids have codon preferences. A system analysis was performed to detect the location and distribution of repeat sequences, and around 197 SSR loci and 51 long repeat sequences were identified in *E. ophiuroides* cp genome. Comparative genomic analysis revealed that *E. ophiuroides* has a high level of collinearity with the other Gramineae cp genomes. Phylogenetic analyses authenticated the close relationship among *E. ophiuroides*, *E. ciliaris*, and *E. eriopoda.* The three Eremochloa species, together with *Mnesithea helferi*, were placed in a monophyletic group corresponding to the subtribe Rottboelliinae, which is completely in accord with the traditional morphology-based taxa of Rottboelliinae in the tribe Andropogoneae. The cp genome information of *E. ophiuroides* could be utilized for species identification, taxonomic clarification, and phylogenetic resolution. Some species-specific markers can be identified and included in this study.

**Supplementary Materials:** The following supporting information can be downloaded at: https://www.mdpi.com/article/10.3390/cimb46020106/s1, Table S1: Statistics of codon preference in cp protein-coding genes of the *E. ophiuroides*, Table S2: Statistics of forward repeats, palindromic repeats and reverse repeats in cp genome of the *E. ophiuroides*, Table S3: SSRs distribution in cp genome of the *E. ophiuroides*, Table S4: Species information and GenBank accession number of the sequenced cp genome used in this study, Table S5: Information of sequenced cp genomes of Panicoideae, Chloridoideae, Oryzoideae, and Pooideae.

**Author Contributions:** Conceptualization, J.L. (Jianxiu Liu) and J.L. (Jianjian Li); methodology, J.L. (Jianjian Li); software, H.W. and J.L. (Jianjian Li); formal analysis, J.Z. and H.W.; investigation, Y.Z., L.Z., J.W., H.G., J.Z., J.C., D.L., and L.L.; resources, J.L. (Jianxiu Liu) and J.L. (Jianjian Li); data curation, H.W., J.W., H.G., J.C., D.L., and L.L.; writing—original draft preparation, J.L. (Jianjian Li); writing—review and editing, H.W., J.L. (Jianxiu Liu) and J.L. (Jianjian Li). All authors have read and agreed to the published version of the manuscript.

**Funding:** This research was funded by the National Natural Science Foundation of China (Grant No. 32072608, 32371767, 31902046, and 32102424), the Jiangsu Agricultural Science and Technology Independent Innovation Fund (Grant No. CX (22) 3175), and the Natural Science Foundation of Jiangsu Province (Grant No. BK20210162).

**Institutional Review Board Statement:** Not applicable.

**Informed Consent Statement:** Not applicable.

**Data Availability Statement:** Sequence information of the newly sequenced *E. ophiuroides* cp genome is available in the NCBI database (www.ncbi.nlm.nih.gov/, accessed on 18 July 2021) under the accession number MT806102. The reference cp genome of *E. ophiuroides* and the whole cp genome sequences of 48 species analyzed in this study were all downloaded from the NCBI database (www.ncbi.nlm.nih.gov/, accessed on 25 July 2021) with their accession numbers listed in Table S1.

**Acknowledgments:** The authors would like to acknowledge Aigui Guo for helping with the maintenance of centipedegrass plants.

**Conflicts of Interest:** The authors declare no conflicts of interest.

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
