# Peer review of "Molecular Characterization and Phylogenetic Analysis of Centipedegrass [Eremochloa ophiuroides (Munro) Hack.] Based on the Complete Chloroplast Genome Sequence"

_cimb, doi:10.3390/cimb46020106_

Round 1

Reviewer 1 Report

Comments and Suggestions for Authors

Reviewer’s Comment / Report

The manuscript #cimb-2838732 entitled “Molecular characterization and phylogenetic analysis of centipede grass [Eremochloa ophiuroides (Munro) Hack.] based on complete chloroplast genome sequence." has been reviewed.

The authors have done significant work on sequencing of the E. ophiuroides cp genome and comparative analysis with other Gramineae related species.

This manuscript reveals that the study assembled and annotated the complete cp genome of E. ophiuroides, which is novelty. The following are some concerns that need to be resolved.

Species names must be corrected italic throughout the manuscript and supplementary files.

The manuscript concluded that the distribution of repeat sequences, and around 197 SSR loci and 51 repeat sequences were 391 identified in E. ophiuroides cp genome.

In methods, Mention the interruptions bases between two adjacent microsatellites to specify the compound microsatellite type.

The author concludes that the cp genome could be used for species identification, taxonomic clarification, and phylogenetic resolution.

Some species-specific markers can be identified and included with this study

Author Response

To Reviewer #1:

  1. Species names must be corrected italic throughout the manuscript and supplementary files.

Response: Thanks for the reviewer’s reminder. We have checked and italicized all Latin names of the species throughout the MS and supplementary files, and also corrected them one by one.

  1. The manuscript concluded that the distribution of repeat sequences, and around 197 SSR loci and 51 repeat sequences were 391 identified in ophiuroides cp genome.

Response: Thanks for the reviewer’s comment. For the imprecise description of the different types of repetitive sequences in original MS, we have revised the corresponding description in the revised MS highlighted.

  1. In methods, Mention the interruptions bases between two adjacent microsatellites to specify the compound microsatellite type.

Response: We have supplemented the corresponding content in materials section 4.3 of the revised MS as shown in highlight.

  1. The author concludes that the cp genome could be used for species identification, taxonomic clarification, and phylogenetic resolution.

Response: We have revised the conclusion section in revised MS based on the comments of the reviewer.

  1. Some species-specific markers can be identified and included with this study

Response: Thanks so much for the insightful suggestions of the reviewer. Since the main work of this MS was mainly focused on molecular characterization of E. ophiuroides cp and phylogenetic analyses, we just mined different types of repeat sequences in this study. And for our further research, we will screen E. ophiuroides specific markers so as for species identification and marker assisted selection in E. ophiuroides.

Reviewer 2 Report

Comments and Suggestions for Authors

Dear Authors,

Thank you very much for your good manuscript. It is well-written and presents novelty results. However, some needed information should be added to the manuscript for the effectiveness of the manuscript.

The manuscript which is entitled “Molecular characterization and phylogenetic analysis of centipedegrass [Eremochloa ophiuroides (Munro) Hack.] based on complete chloroplast genome sequence” shows novel and original results. The manuscript reveals assembled the complete cp genome sequence of E. ophiuroides, characterized the cp genomic structure and performed detailed phylogenetic analyses using complete cp genome sequence information. Also the study lead next studies for the studied plant.

There are some missing information in the manuscript as follows:

1- Line 16-17: Abstract Part: Replace the this sentence, importance of the studied plant should be given. Also this sentence can be added at the end of the introduction part for the emphasize the study.

2. The figures as figure 1 or others can be presented as more clearly.

3. The discussion part should be developed, and comparisons should be made with previous studies.

4. The "Plant material, DNA extraction and sequencing" in Material and Methods section should be written as detailed.

Best regards,

Author Response

To Reviewer #2:

  1. Line 16-17: Abstract Part: Replace the this sentence, importance of the studied plant should be given. Also this sentence can be added at the end of the introduction part for the emphasize the study.

Response: We have rewritten the corresponding sentence in Abstract part and also added the related contents in Introduction section in the revised MS as highlighted.

  1. The figures as figure 1 or others can be presented as more clearly.

Response: The clear figure files can be found in the submitted “Figures-chloroplast-v1.zip” file, named “Fig.1.pptx ~ Fig.5.pptx”.

  1. The discussion part should be developed, and comparisons should be made with previous studies.

Response: For discussion part in our MS, whether the chloroplast genome size and gene number, repeat sequences composition, IR expansion and contraction, or phylogentic analysis, we have compared and discussed them in detail with the results of previous studies point-by-point,   Therefore, in order to keep the conciseness and coherence of the MS, we prefer to maintain the existing status.

  1. The "Plant material, DNA extraction and sequencing" in Material and Methods section should be written as detailed.

Response: According to the reviewer’s comment, we have supplemented the corresponding content for "Plant material, DNA extraction and sequencing" section as shown in highlight in the revised MS.

Reviewer 3 Report

Comments and Suggestions for Authors

Eremochloa ophiuroides provides a good model for studying the chloroplast genome and comparing it with those already known from this large family. In addition, the E. ophiuroides genome can be used in the future to identify mechanisms of resistance and adaptation to many abiotic and biotic stresses. The high development of technologies in genome sequencing and the high qualifications of the authors in deciphering the data obtained allowed them to present such a high-quality and interesting manuscript. The manuscript is carefully prepared, so there are few comments.

141- remove the polyA decoding, etc., since these are common abbreviations

149 - add "Gramineae or Poaceae"

213- why "aproximately" if the exact bp number is given

259- A/T give as -99 (AT)

279 - clarify the origin of C4 plants, as well as -286 C3

342 - de novo - italics

Author Response

To Reviewer #3:

  1. 141- remove the polyA decoding, etc., since these are common abbreviations.

Response: We have omitted them in revised MS.

  1. 149 - add "Gramineae or Poaceae"

Response: We have replaced "Gramineae” with "Gramineae or Poaceae" in revised MS as shown in highlight.

  1. 213- why "aproximately" if the exact bp number is given.

Response: Thanks for the reviewer’s reminder. We have deleted “approximately” in line 213 in revised MS.

  1. 259- A/T give as -99 (AT)

Response: We have omitted oblique line in line 259 in revised MS.

  1. 279 - clarify the origin of C4 plants, as well as -286 C3

Response: We have supplemented the corresponding origins of C4 and C3 plants as marked in highlight in the revised MS.

  1. 342 - de novo - italics

Response: We have italicized it in the revised MS.